# Blood-Based Immune Profiling Combined with Machine Learning Discriminates Psoriatic Arthritis from Psoriasis Patients

**DOI:** 10.3390/ijms222010990

**Published:** 2021-10-12

**Authors:** Michelle L. M. Mulder, Xuehui He, Juul M. P. A. van den Reek, Paulo C. M. Urbano, Charlotte Kaffa, Xinhui Wang, Bram van Cranenbroek, Esther van Rijssen, Frank H. J. van den Hoogen, Irma Joosten, Wynand Alkema, Elke M. G. J. de Jong, Ruben L. Smeets, Mark H. Wenink, Hans J. P. M. Koenen

**Affiliations:** 1Department of Rheumatology, Sint Maartenskliniek, 6524 Nijmegen, The Netherlands; m.mulder@maartenskliniek.nl (M.L.M.M.); fr.vandenhoogen@maartenskliniek.nl (F.H.J.v.d.H.); m.wenink@maartenskliniek.nl (M.H.W.); 2Department of Dermatology, Radboud University Medical Center, 6524 Nijmegen, The Netherlands; juul.vandenreek@radboudumc.nl (J.M.P.A.v.d.R.); elke.dejong@radboudumc.nl (E.M.G.J.d.J.); 3Department of Laboratory Medicine—Medical Immunology, Department of Dermatology, Radboud University Medical Center, 6524 Nijmegen, The Netherlands; xuehui.he@radboudumc.nl (X.H.); Paulourban86@gmail.com (P.C.M.U.); bram.vancranenbroek@radboudumc.nl (B.v.C.); esther.vanrijssen@radboudumc.nl (E.v.R.); irma.joosten@radboudumc.nl (I.J.); ruben.smeets@radboudumc.nl (R.L.S.); 4Center for Molecular and Biomolecular Informatics, Radboud University Medical Center, 6524 Nijmegen, The Netherlands; charlotte.kaffa@radboudumc.nl; 5Luxembourg Centre for Systems Biomedicine, University of Luxembourg, L-4475 Belvaux, Luxembourg; xinhui_wang@hotmail.com; 6College of Computer Science, Qinghai Normal University, Xining 810000, China; 7Institute for Life Science and Technology, Hanze University of Applied Sciences, 9727 Groningen, The Netherlands; wynand.alkema@tenwise.nl; 8TenWise BV, 5344 KX Oss, The Netherlands; 9Department of Laboratory Medicine, Laboratory for Diagnostics, Radboud University Medical Center, 6524 Nijmegen, The Netherlands

**Keywords:** psoriatic arthritis, psoriasis, detection, immune profile, flow cytometry, machine learning

## Abstract

Psoriasis (Pso) is a chronic inflammatory skin disease, and up to 30% of Pso patients develop psoriatic arthritis (PsA), which can lead to irreversible joint damage. Early detection of PsA in Pso patients is crucial for timely treatment but difficult for dermatologists to implement. We, therefore, aimed to find disease-specific immune profiles, discriminating Pso from PsA patients, possibly facilitating the correct identification of Pso patients in need of referral to a rheumatology clinic. The phenotypes of peripheral blood immune cells of consecutive Pso and PsA patients were analyzed, and disease-specific immune profiles were identified via a machine learning approach. This approach resulted in a random forest classification model capable of distinguishing PsA from Pso (mean AUC = 0.95). Key PsA-classifying cell subsets selected included increased proportions of differentiated CD4+CD196+CD183-CD194+ and CD4+CD196-CD183-CD194+ T-cells and reduced proportions of CD196+ and CD197+ monocytes, memory CD4+ and CD8+ T-cell subsets and CD4+ regulatory T-cells. Within PsA, joint scores showed an association with memory CD8+CD45RA-CD197- effector T-cells and CD197+ monocytes. To conclude, through the integration of in-depth flow cytometry and machine learning, we identified an immune cell profile discriminating PsA from Pso. This immune profile may aid in timely diagnosing PsA in Pso.

## 1. Introduction

Psoriasis (Pso) is a chronic and common skin disease with a prevalence of around 2% [1]. Significant co-morbidities are associated with Pso, including psoriatic arthritis. Up to 30% of Pso patients develop psoriatic arthritis (PsA), a chronic inflammation of joints and entheses with a lag time of about 10 years. PsA has an additional negative impact on quality of life and delayed diagnosis leads to increased chances of disability [1,2,3,4]. Adequate detection of PsA in Pso patients is, therefore, important to ensure timely treatment and prevention of structural joint damage [5]. However, identifying patients with (concomitant) PsA in psoriasis patients is difficult among others due to the diverse presentation of the disease. There is an unmet need for a valid method that is able to assist in screening Pso patients for PsA.

Considering both diseases are immune-mediated inflammatory diseases but with clear differences in clinical characteristics, it is conceivable that these differences may be reflected in the composition of circulating immune cell subsets. Several serum soluble biomarkers, such as CXCL10, matrix metalloproteinase-3 (MMP-3) and macrophage-colony stimulating factor (M-CSF) have been shown to be associated with PsA [6,7]. Additionally, increased percentages of CD8+ effector T-cells were reported in PsA [8]. However, to our knowledge, a biomarker (or a subset of markers) with enough discriminative capacity to use in routine (dermatology) practice to identify PsA patients in a cohort of Pso patients is still lacking.

Here, we performed a cross-sectional study to investigate whether PsA patients can be distinguished from Pso patients by integrating flow cytometry of circulating peripheral blood immune cell subsets and a machine learning approach. We identified distinguishing, disease-specific immune profiles, which might aid in the early diagnosis of Pso patients with concomitant PsA.

## 2. Results

### 2.1. Patient Population

The study consecutively enrolled 41 PsA and 45 Pso patients. Extensive information on patients and disease characteristics can be found in Table 1. The percentage of female patients was 53.7% and 44.4% in the PsA group and Pso group, respectively. Mean age was, as expected, higher in PsA patients (56.1 ± 14.5 years) compared to Pso patients (44.1 ± 15.2 years). All the PsA patients had peripheral disease and did not have predominant axial disease. None of the included patients had concomitant other immune-mediated inflammatory diseases (e.g., inflammatory bowel disease).

### 2.2. Differences in Immune Cell Subsets between PsA and Pso

Deep phenotyping of circulating immune cell subsets resulted in the detection of over a hundred different cell subsets for each blood sample. Univariate analysis of cell percentages data revealed a number of immune cell subsets including T, B, NK, NKT and monocytes that were significantly different in PsA vs Pso (Figure 1A). As compared to Pso, PsA patients showed reduced levels of CD196+ and CD197+ monocytes, memory T-cells, CD4+CD197+CD45RA- T-cells, CD8+CD45RA-CD27- T-cells, Treg, and CD19+IgD+CD5++ B-cells, but increased fractions of CD4+CD196+CD183-CD194+ Th17-like and CD4+CD196-CD183-CD194+ T-cells (Figure 1B). Details on cell subsets analyzed in the univariate analysis are shown in Appendix A.

### 2.3. Random Forest Classification Model Reveals a Disease-Specific Immune Profile

To determine whether Pso could be distinguished from PsA by their immune cell phenotype, a machine learning classification model using Random Forest (RF) methodology was generated by using selected immune cell subsets that were shown to be significantly different between PsA and Pso and that were not highly correlated with each other (Spearman rho < 0.8). The analytical procedure is illustrated in Figure 2A. The resulting ROC curve of 500 RF models indicates a proper discrimination between PsA and Pso (Figure 2B, mean AUC value 0.95 ± 0.04; mean specificity 0.73 ± 0.04; mean sensitivity 0.71 ± 0.04; mean out-of-bag error 0.13). Age did not influence the classification model performance (mean AUC value 0.95 ± 0.04; mean specificity 0.73 ± 0.04; mean sensitivity 0.71 ± 0.04).

To analyze to what extent the different immune cell subsets contributed to the classification of PsA and Pso, we used the feature importance score (gini score) generated by the RF classification model. Figure 2C lists the top 10 immune cell subsets based on their gini scores, which quantitatively measures how on average a variable contributes to the model. The higher the gini score, the more important the feature is for the classification. The most relevant cell subsets contributing to the classification of PsA or Pso were memory T-cells (CD4+CD197+CD45RA-, CD45RO+CD45RA-, CD8+CD45RA-CD27-, CD4+CD45RA-CD27-), Th17-like cells (CD4+CD196+CD183-CD194+), Th2-like cells (CD4+CD196-CD183-CD194+), and Treg cells (CD4+CD25+CD127lo). Additionally, CD196+ and CD197+ monocytes helped to distinguish PsA from Pso (Figure 2C).

### 2.4. Association of the PsA-Specific Immune Profile with Clinical Parameters

We applied a qualitative analysis to assess the correlations between disease activity measures and the percentages of the top 10 PsA/Pso-classifying immune cell populations. The correlation of cell percentages with joint disease activity (swollen joint count (SJC) 28/66, tender joint count (TJC) 28/68), disease activity composite scores (Disease Activity Score 28 (DAS28) and Psoriatic Arthritis Disease Activity Score (PASDAS)) and skin disease severity (Psoriasis Area and Severity Index (PASI)), in the PsA cohort, was assessed. Decreased numbers of CD197+ monocytes and CD8+CD45RA-CD27- memory T-cells positively correlated with joint involvement as well as overall disease composite scores (Figure 3). None of the PsA/Pso-classifying immune cell subsets showed a correlation with skin disease severity (PASI) in PsA, nor in Pso, except for CD196+ monocytes which mildly correlated with PASI in Pso (r = 0.38, *p* < 0.05) (Figure 4).

## 3. Discussion

To our knowledge, this is the first study to show that PsA and Pso patients can be discriminated based on blood-based profiling of multiple immune cell subsets combined with a machine learning approach. The PsA-specific immune profile distinguishing PsA from Pso, is defined by a reduced proportion of CD4 and CD8 memory T-cell subsets, Treg cells and CD196+ and CD197+ monocytes as well as an increased proportion of differentiated CD4+ memory T-cells expressing the chemokine receptors CD196 (CCR6) and CD194 (CCR4).

The reduced presence of memory T cells and CD196 (CCR6)+ monocytes in PsA patients′ peripheral blood as compared to Pso patients might be explained by increased migration to the joints and entheses of the PsA patients. The increased proportion of memory CD4+ T-cells expressing the chemokine receptors CD194 (CCR4) and CD196 (CCR6) in the circulation of PsA patients might appear contradictory to this. However, memory T cells expressing CD196 (CCR6) have been shown to be increased in the peripheral blood of PsA patients before while they also accumulated in the synovial cavity and migrated towards PsA synovial fluid [8]. These results hint at the specifically increased proliferation of CD196 (CCR6)+ (memory) T cells in PsA compensating for the accumulation in the joint/entheses.

CD194 (CCR4)+ T cells have also been shown to migrate to the PsA synovial cavity, which expresses high levels of the CD194 (CCR4) ligand MDC/CCL22 [9]. In line, Th17 and Th22 cell populations are enriched for CD196 (CCR6) and CD194 (CCR4) and are known to play a pathogenic role in the pathogenesis of PsA [10,11]. As CD196 (CCR6)+ and CD194 (CCR4)+ (memory) CD4+ T cells are also implicated in Pso, there might also be a relative decrease of these immune cell subsets in the peripheral blood of Pso patients e.g., due to the increased inflammation of the skin in the Pso vs PsA in our populations. However, we did not find any correlation between PASI scores and the CD196 (CCR6)+ and CD194 (CCR4)+ (memory) CD4+ T cell subset (Figure 4) making it less likely that this effect plays a major role.

In PsA patients we further observed fewer circulating CD197 (CCR7)+ monocytes as compared to Pso and this cell population revealed a strong correlation with joint disease activity as well as overall disease composite scores, suggesting that CD197 (CCR7)+ monocytes might migrate to the lymphatic system and/or joints and entheses and contribute to inflammation. Others have reported that CD197 (CCR7) mRNA levels in monocytes correlate to the clinical inflammation status (DAS28) in rheumatoid arthritis patients and that signaling of CD197 (CCR7) (by CCL21) promotes Th17-mediated bone loss an important destructive process in PsA [12]. In contrast, in models of psoriasis, no role for CD197 (CCR7)+ monocytes/monocyte-derived dendritic cells was found in the disease process [13]. Noteworthy are the findings by Schnitte et al., who found that peripheral blood monocyte-derived dendritic cells from PsA patients migrated less towards a MIP3b gradient (CCR7 ligand) than did cells from Pso patients while the level of expression of CD197 (CCR7) was comparable [14]. They did not report if a difference in numbers of CD197 (CCR7)+ monocytes/monocyte-like cells was researched. These data might be in line with our hypothesis in which migration-prone CD197 (CCR7)+ cells have already migrated to lymphatics and sites of inflammation and the less responsive and/or MIP3b pre-exposed cells remain.

The strengths of this study are (1) extensive phenotyping of more than one hundred immune cell subsets, (2) unsupervised machine learning data analysis approach that minimizes bias and strengthens its robustness, and (3) availability of extensive clinical data enabling correlation of PsA-classifying immune cell subsets with clinical parameters. This resulted in a capability to distinguish Pso from PsA patients. Further validation of our results is found in the correlation of the PsA-identifying cell subsets with the joint scores and musculoskeletal disease activity measures within the PsA population.

With regard to the limitations of our study, we found differences in three important clinical parameters between PsA and Pso patients. First, the difference in PASI scores between the PsA and Pso cohorts might be a limitation of our study. However, we deem it unlikely that our findings are influenced by the difference in PASI score, as no clear correlation between (available) PASI scores and the PsA-classifying immune cell populations were found. Second, a difference in conventional disease-modifying antirheumatic drug (cDMARD) use was present at baseline. The cDMARD use in PsA patients was higher than in Pso patients. To increase the external generalizability of our results, we did not exclude patients that used a cDMARD. As our capability to distinguish PsA from Pso patients was high, the difference in the use of cDMARDs seems unlikely to play a decisive role. Last, a difference was found in the age of PsA and Pso patients, with the PsA patients (56.1 ± 14.5) being about 10 years older than the Pso patients (44.1 ± 15.2) but also with a large age overlap. As we took consecutive patients from both cohorts, this difference in age was expected [15]. In fact, including the patients’ age did not improve the RF model performance. Age has been evaluated extensively in the past also using the same flow cytometry panels used in the current study [16,17]. Although aging influences the immune cell composition (e.g., young vs old), within the age-interval (mean age 44–56) of our study cohort, as expected and as analyzed, the outcome of the RF model was hardly influenced.

However, although an important role played by the above variables is unlikely, we aim to conduct a longitudinal follow-up study in which we match Pso patients with newly found PsA patients from the same cohort. In our current work, we used five supplemental 10-color antibody panels to reveal an immune profile that discriminates between PsA and Pso. As this is not suitable for daily clinical practice, in future studies, based on our RF-modelling, only a single panel might be used. A panel consisting of the immune lineage markers CD3, CD4, CD8, CD14, combined with the maturation/differentiation markers CD45RA, CD27, CD127, activation marker CD25 and the chemokine receptors CD196, CD197, CD183 as this appears to be sufficient to discriminate Pso patients with and without PsA.

To conclude, our study shows that Pso and PsA patients can potentially be differentiated based on the flow cytometry of circulating immune cells combined with a machine learning approach. This highlights the possibility, after external validation, of using a combination of selected immune cell subsets, as a method for detecting PsA in Pso patients.

## 4. Materials and Methods

### 4.1. Patients and Setting

Patient inclusion criteria were (1) a diagnosis of Pso by a dermatologist or PsA made by a rheumatologist, respectively, and (2) age ≥ 18 years. Exclusion criteria consisted of (1) current malignancies and (2) the presence of other inflammatory rheumatic diseases. The Pso cohort included patients with only cutaneous involvement; all Pso patients were screened for PsA by using the Psoriasis Epidemiology Screening Tool (PEST) (a PEST score of ≥2 was used as cut-off value), or concomitant PsA was ruled out after assessment by a rheumatologist [18]. Demographic and clinical data were collected. PASI was used to measure skin disease severity. Joint disease activity was assessed using both the 28 and 66/68 tender and swollen count (TJC/SJC). Disease activity composite scores in PsA were assessed using the DAS28 and the PASDAS. Ethical approval (number: NL66544.091.18) or exemption of formal ethical review was obtained from the local human research ethics committee (Radboudumc, Nijmegen, The Netherlands).

### 4.2. Experimental Procedure

Peripheral blood (PB) samples of Pso and PsA patients were collected in 10 mL K2EDTA tubes, processed for flow cytometry and analyzed as reported [19]. Flow cytometry panels and antibodies are listed in Appendix A. All antibodies were titrated and tested before use. Percentages of immune cell subsets were manually analyzed using Kaluza software version 2.1 (Beckman-Coulter, Brea, CA, USA) as described previously [16,17,19].

### 4.3. Statistical Analysis

Subsets that were shown to be significantly different (Wilcoxon signed-rank test) between PsA and Pso and that were not highly correlated (spearman rho < 0.8), were included as features in the RF classification model. To build the RF classifier, the following steps were taken (Figure 2A). (1) The dataset was randomly split into a training (70% of the samples) and test-set (30% of the samples). (2) An RF model consisting of 1000 forests was built and optimized with internal cross-validation. (3) Steps 1 and 2 were repeated 500×. This procedure resulted in 500 RF-based models, each of which was evaluated by an Area under the curve (AUC) analysis. Pearson′s correlation coefficient was used to assess the correlations of immune cell subsets and clinical parameters. All data were analyzed using the statistical packages installed under R version 3.6.2 (https://www.r-project.org, accessed on 19 September 2021).

## Figures and Tables

**Figure 1 ijms-22-10990-f001:**
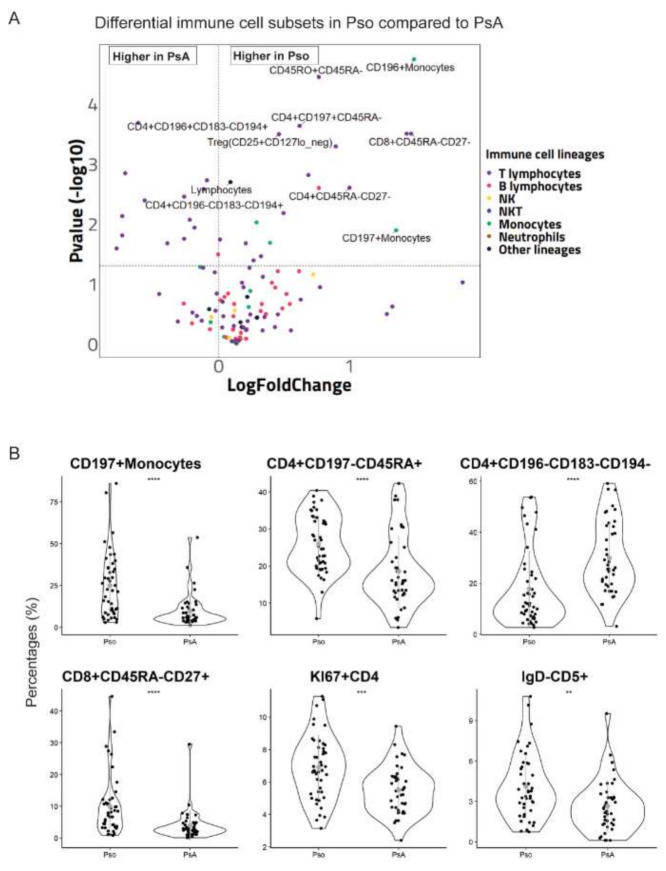
Differences in the percentages of circulating immune cell subsets in PsA versus Pso. (**A**) The volcano plot shows the differential immune cell subsets in Pso versus PsA patients. Each dot represents one cell subset and is colored based on its lymphocytes lineage as indicated in the legend. The X-axis indicates the log2 (fold change of Pso/PsA). The horizontal dotted line indicates *p* = 0.05. The vertical dotted line indicates Pso/PsA fold change = 1. (**B**) The violin plots show examples of significantly different cell subsets in PsA versus Pso patients. Each dot represents an individual subject. The grey dot and grey line indicate the mean values and two times standard deviations, respectively.

**Figure 2 ijms-22-10990-f002:**
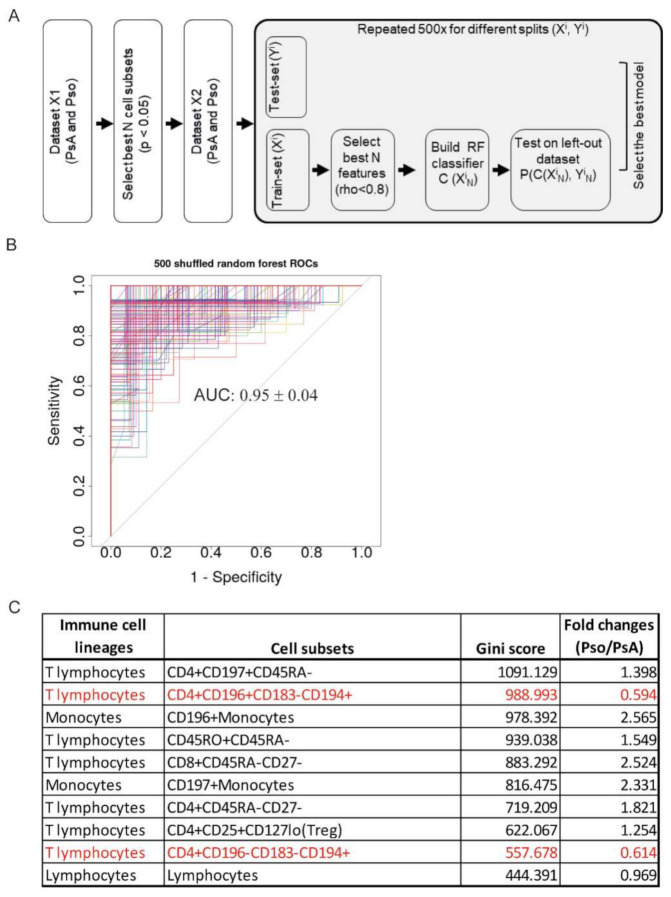
Discrimination of PsA from Pso patients using a random forest classification model. (**A**) Schematic overview of the data analysis procedure. The significantly different cell subsets in PsA versus Pso were selected based on the univariate analysis as shown in Fig 1A and the highly correlated cell subsets were excluded. Randomly splitting for training X’ (70%) and test Y’ (30%) dataset was repeated 500×. The RF classification model (containing 1000 forests) was built using the training dataset and the test dataset was shuffled randomly to cross-validate the model. (**B**) Overview of the ROC curves derived from 500 RF classification models in which selected immune cell subsets (*p*-value < 0.05 and rho < 0.8) function as predicting variables. The AUC value was shown as mean ± SD. (**C**) Top 10 most relevant-cell subsets contributing to the classifications of PsA and Pso. Cell subsets were ranked based on gini-score. Fold change is computed as the ratio of mean values of each cell subset’s percentage in Pso vs PsA. Names of immune cell subsets in red indicate these cells are higher in PsA than in Pso.

**Figure 3 ijms-22-10990-f003:**
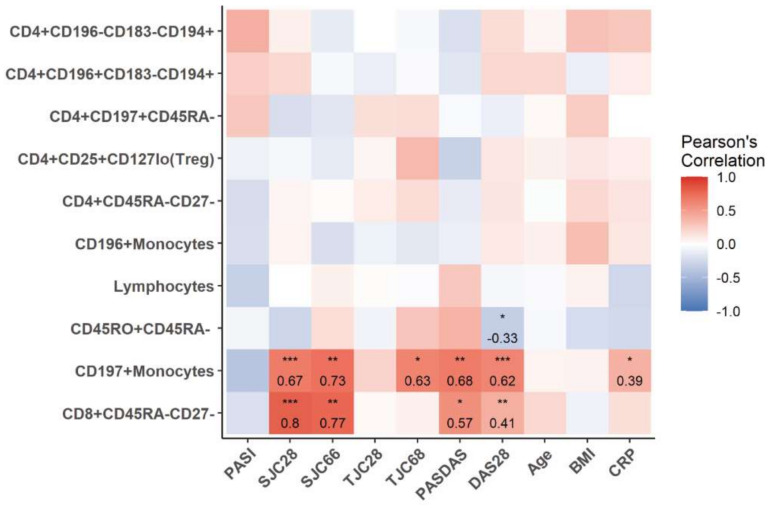
Correlation of clinical parameters with key immune cell subsets that contribute to the discrimination of PsA and Pso. ^1^ Heatmap shows the Pearson′s correlation between percentages of top 10 PsA-classifying cell subsets with clinical parameters in PsA. Numbers indicate the Pearson’s correlation coefficient, * *p* < 0.05; ** *p* < 0.01; *** *p* < 0.001.

**Figure 4 ijms-22-10990-f004:**
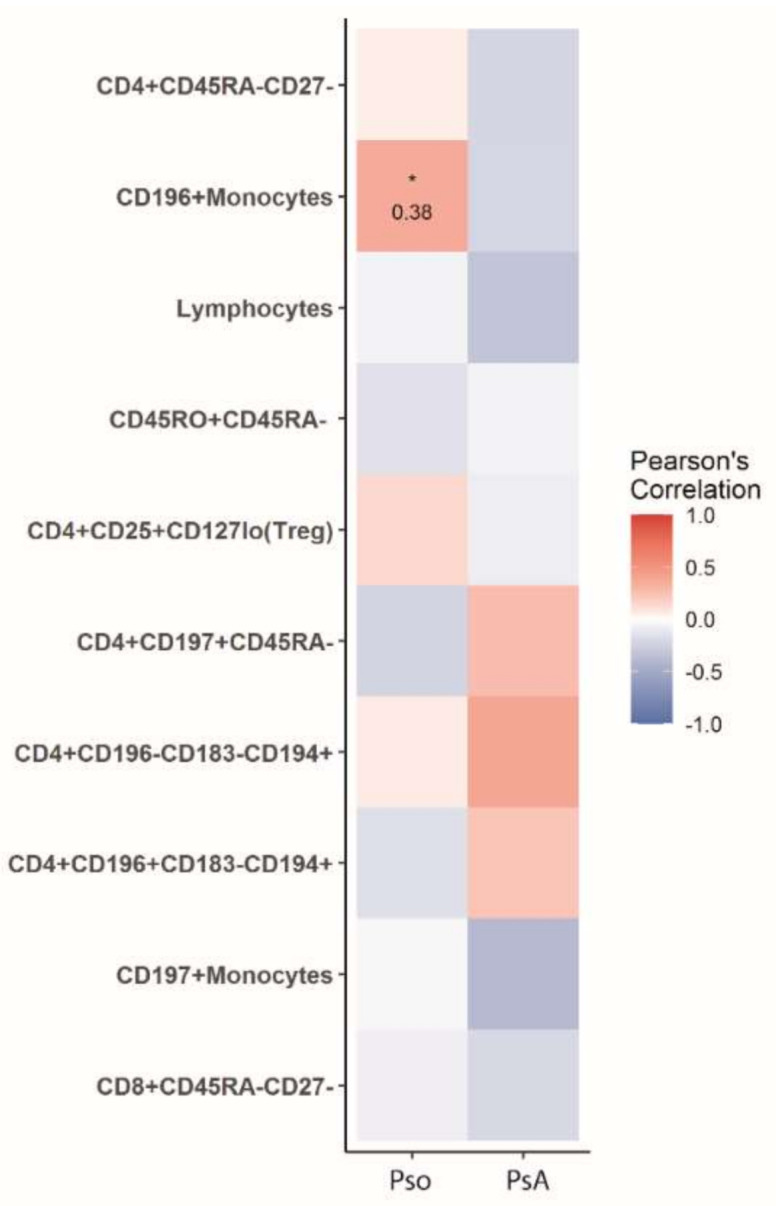
Correlation of PASI with key immune cell subsets that contribute to the discrimination of PsA and Pso. ^1^ Heatmap shows the Pearson′s correlation between percentages of top 10 PsA-classifying cell subsets with PASI score. Numbers indicate the Pearson′s correlation coefficient. * *p* < 0.05.

**Table 1 ijms-22-10990-t001:** Demographic and clinical characteristics of Pso and PsA patients.

	PsA (*N* = 41)	Pso (*N* = 45)
Age (years)	56.1 ± 14.5	44.1 ± 15.2
Female (number, %)	22 (53.7%)	20 (44.4%)
BMI	27.0 ± 5.1 ^#^	29.2 ± 5.4 ^#^
cDMARD (current use)	17 (41.5%)	4 (8.7%)
bDMARD (current use)	0 (0%)	7 (15.6%)
CRP	4.8 ± 10.7	3.2 ± 5.0 ^##^
PASI	2.9 ± 3.7 ^###^	13.3 ± 7.3
DAS28	2.4 ± 1.3	-
PASDAS	4.9 ± 1.1 ^###^	-
TJC28	2.2 ± 3.6	-
TJC68	7.2 ± 5.8 ^###^	-
SJC28	1.8 ± 3.5	-
SJC66	5.8 ± 6.4^###^	-

^1^ Except where indicated otherwise, values are mean ± SD. bDMARD = biological disease-modifying antirheumatic drug; BMI = body mass index; cDMARD = conventional DMARD; CRP = C-reactive protein; DAS28 = disease activity score in 28 joints (range 0.96–10) PASI = psoriasis area and severity index (range 0–72); PASDAS = psoriatic arthritis disease activity score (range 0–10); SJC28 = Swollen joint count of 28 joints (range 0–28); SJC66 = Swollen joint count of 66 joints (range 0–66); TJC28 = Tender joint count of 28 joints (range 0–28); TJC68 = Tender joint count of 68 joints (range 0–68). ^2^ Variables with missing values for PsA and/or Pso. ^#^ BMI was available for a subset of 27 PsA patients and 41 Pso patients.^##^ CRP was available for a subset of 42 Pso Patients. ^###^ Data was available for a subset of 14 PsA patients.

## Data Availability

The data underlying this article will be shared on reasonable request to the corresponding author.

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
