# Peer review of "Blood-Based Immune Profiling Combined with Machine Learning Discriminates Psoriatic Arthritis from Psoriasis Patients"

_ijms, 2021, doi:10.3390/ijms222010990_

Round 1

Reviewer 1 Report

Authors have claimed to find disease-specific immune profiles, discriminating Pso from PsA patients using the Random Forest Technique. From the Data science viewpoint, authors have used a standard ML pipeline and RF as prediction techniques. To find the subset of prominent features, the Gini index has been used. 
As indicated by the authors, the strength of this study lies in:
1) extensive phenotyping of more than one hundred 211 immune cell subsets, 
2) unsupervised machine learning data analysis approach that 212 minimizes bias and strengthens its robustness.
3) availability of extensive clinical data 213 enabling correlation of PsA-classifying immune cell subsets with clinical parameters.
In my capacity as ML practitioner, I did not find much novelty/experimentation depth regarding item no. 2 above; However, efforts involved or modernity of the work included in items 1 and 3 above may be reviewed by a biologist. However, I feel that Random forest has been used appropriately. 

Author Response

We thank reviewer 1 for recognizing the strengths of the study and appropriateness of analytical methods. We do agree with reviewer 1 that the use of a random forest model in general isn’t very novel. However, when looking at this specific research question (distinguishing Pso from PsA patients), we do believe it is a novel approach, as we could not find any literature regarding the use of the RF model for this purpose. Furthermore, we believe that the amount of immune cell subsets that we looked into and identified is much larger than usually reported in other studies. Likewise, the amount of clinical variables is relatively extensive. Because also clinicians were involved in -among others- collecting the data and writing the manuscript (dermatologists and rheumatologists), correlation between relevant clinical variables and immune cell subsets forms a substantial part of our manuscript.

Reviewer 2 Report

The manuscript by Mulder et al. describes the identification of a specific profile of circulating immune cells that helps distinguish patients with Psoriasis (Pso) from patients that have psoriatic arthritis (PsA). As the authors indicate, there is an unmet need for a method to identify Pso patients in need of referral to a rheumatologist because they will develop PsA, in particular in view that a late diagnosis results in further disease complications and reduced quality of life. The specific profile was obtained by using a machine learning approach with random forest classification model on data obtained from in-depth flow cytometry characterisation of peripheral blood immune cells.

The manuscript is clear and concise, the methods are state-of the art and the conclusions are nicely drawn from the experimental evidence.

I have a single concern. The immune profiling described here is very useful to distinguish Pso from PsA patients. However, the unmet need is for a method that identifies Pso patients in need of referral to a rheumatologist because they will develop arthritis (i.e. not yet a full-blown PsA patient). Do the authors think that this is feasible on the basis of the subpopulations of immune cells identified in both groups of patients? Which particular cell types should be watched, in light of what we know about the pathologic process in PsA?

Author Response

We would like to thank reviewer 2 for appreciating the relevance of this study and his/her positive feedback.

Reviewer 2 raises a valid point with regard to the patient cohorts that were used in this study. To answer the clinically very relevant research question proposed by reviewer 2 correctly (identifying sub-clinical PsA patients in a psoriasis cohort), we would have to perform a longitudinal study following patients with Pso until they develop PsA. We clarified this in the discussion section by adding the word “longitudinal” to the following sentence (lines 236-238):

“However, although an important role played by the above variables is unlikely we aim to conduct a longitudinal follow-up study in which we match Pso patients with newly found PsA patients from the same cohort.”

Because conversion from Pso to PsA usually takes a substantial amount of time, we chose to first get an indication whether this analytical method was able to differentiate between Pso and PsA by performing this cross-sectional study. We believe that because of the good performance of the model, it might indeed be promising for identifying patients with (very) early PsA. This, of course, has to be confirmed in additional validation studies. But as the majority of the PsA patients had rather low disease activity, this does indicate that this analytical method is also able to pick-up PsA patients without clinical active disease.

The question whether “specific immune cell subsets we found capable of distinguishing PsA from Pso patients are the most essential subsets to find subclinical PsA patients in a Pso cohort” is harder to answer. The fact that as stated above disease activity in the PsA patients was relatively low and certain cell subsets clearly correlate with joint involvement/composite scores suggests that these subsets might already start to differ at the earliest stages in the disease process. In addition CCR6+ Th(17) cells have been implicated widely in the disease process of PsA as we described in the discussion, so these might be interesting to look into in the earliest phases of the development of PsA. Furthermore, CCR7+ myeloid cells appear of particular interest in this process since the instruction of adaptive (Th/c-17) immune cells in primary and secondary lymphoid tissue might be an instigating process in the development of chronic inflammation in PsA.

We must however stress that the strength of our study lies in the “unsupervised” analyses of all the different cell subsets via machine learning thus not leaning on any previous knowledge/data. So to hopefully create an equally powerful tool for the detection of subclinical PsA a longitudinal study is preferred with this specific research question.  

Round 2

Reviewer 1 Report

Authors have incorporated all suggested changes in the manuscript.